# Isolation and Characterization of Targeting-HBsAg VNAR Single Domain Antibodies from Whitespotted Bamboo Sharks (*Chiloscyllium plagiosum*)

**DOI:** 10.3390/md21040237

**Published:** 2023-04-12

**Authors:** Xierui Jiang, Shan Sun, Zengpeng Li, Mingliang Chen

**Affiliations:** 1School of Advanced Manufacturing, Fuzhou University, Quanzhou 362251, China; 2Key Laboratory of Marine Genetic Resources, State Key Laboratory Breeding Base of Marine Genetic Resources, Fujian Key Laboratory of Marine Genetic Resources, Fujian Collaborative Innovation Centre for Exploitation and Utilization of Marine Biological Resources, Third Institute of Oceanography Ministry of Natural Resources, Xiamen 361005, China; 3Key Laboratory of Ministry of Education for Conservation and Utilization of Special Biological Resources in the Western, Life Science School, Ningxia University, Yinchuan 750021, China; 4Co-Innovation Center of Jiangsu Marine Bio-Industry Technology, Jiangsu Ocean University, Lianyungang 222005, China

**Keywords:** whitespotted bamboo shark, VNAR, phage display, HBsAg

## Abstract

Immunoglobulin new antigen receptor (IgNAR) is a naturally occurring antibody that consists of only two heavy chains with two independent variable domains. The variable binding domain of IgNAR, called variable new antigen receptor (VNAR), is attractive due to its solubility, thermal stability, and small size. Hepatitis B surface antigen (HBsAg) is a viral capsid protein found on the surface of the Hepatitis B virus (HBV). It appears in the blood of an individual infected with HBV and is widely used as a diagnostic marker for HBV infection. In this study, the whitespotted bamboo sharks (*Chiloscyllium plagiosum*) were immunized with the recombinant HBsAg protein. Peripheral blood leukocytes (PBLs) of immunized bamboo sharks were further isolated and used to construct a VNAR-targeted HBsAg phage display library. The 20 specific VNARs against HBsAg were then isolated by bio-panning and phage ELISA. The 50% of maximal effect (EC_50_) of three nanobodies, including HB14, HB17, and HB18, were 4.864 nM, 4.260 nM, and 8.979 nM, respectively. The Sandwich ELISA assay further showed that these three nanobodies interacted with different epitopes of HBsAg protein. When taken together, our results provide a new possibility for the application of VNAR in HBV diagnosis and also demonstrate the feasibility of using VNAR for medical testing.

## 1. Introduction

Immunoglobulin new antigen receptor (IgNAR) is a class of antibody molecules first isolated from nurse sharks (*Ginglymostoma cirratum*) [1]. It is a unique form of antibody that differs significantly from the commonly known immunoglobulin (Ig) classes, such as IgG, IgM, and IgE, and represents an evolutionary departure from conventional antibody structures [2,3]. The discovery of IgNAR antibodies in shark serum has garnered much attention in the field of immunology due to their unique structural features and potential applications in biotechnology. IgNAR molecules are characterized by a unique antigen-binding domain, which is distinct from those found in conventional antibodies. This domain is composed of two variable heavy (VH) chains that are connected by a disulfide bond, forming a V-shaped structure [4,5]. Each chain consists of five constant domains and a variable domain [6]. The variable domain was called variable new antigen receptor (VNAR) or “shark nanobody.” VNAR antibodies represent a unique form of antibody with distinct structural and functional properties [7,8]. Their stability, specificity, and potential applications in biotechnology make them ideal for diagnostic purposes, where rapid and accurate detection of viruses is crucial. VNARs have been used to detect a range of viral and bacterial pathogens, including the Ebola virus (EBOV), Hepatitis B virus (HBV), and Cholera toxin (CT) [9,10,11]. For example, researchers have developed a diagnostic assay based on VNARs that can detect low levels of the nucleoprotein of the Ebola virus with high sensitivity and specificity [12,13]. This assay has the potential to improve the speed and accuracy of EBOV diagnosis, particularly in outbreaks or pandemics where the rapid and accurate diagnosis is crucial in resource-limited settings. In the case of HBV, VNARs have been used to detect the Hepatitis B e antigen (HBeAg), a marker of the early phase of HBV infection. Researchers have developed a rapid, highly sensitive diagnostic assay based on VNARs that can detect HBeAg in serum samples with high accuracy [14]. This assay has the potential to improve the diagnosis of HBV infection, particularly in resource-limited settings where traditional diagnostic methods may not be readily available. VNARs have also been used to detect Cholera toxin (CT) with high accuracy [15], which improves the speed and accuracy of CT detection in many tropical and sub-tropical developing countries. The use of shark-derived VNARs as a diagnostic tool for viral infections is a promising area of research with numerous applications. The development of rapid, highly sensitive diagnostic assays based on VNARs has the potential to improve the speed and accuracy of virus diagnosis, particularly in resource-limited settings where traditional diagnostic methods may not be readily available.

Hepatitis B virus (HBV) is a hepatophilic DNA virus capable of establishing persistent chronic infections in humans through immune-mediated mechanisms [16]. Currently, 3.9% of the global population is infected with HBV, demonstrating that HBV continues to pose a significant public health challenge, despite the availability of safe and effective vaccines for over four decades [17]. The clinical manifestations of HBV infection are diverse and range from acute hepatitis to various forms of chronic infection, including chronic hepatitis, cirrhosis, liver failure, and hepatocellular carcinoma [18,19]. These conditions highlight the need for continued efforts to control the spread of HBV. The discovery of Hepatitis B surface antigen (HBsAg) in 1968 by Blumberg and colleagues marked a significant milestone in the diagnosis of HBV infection [20]. HBsAg has since been widely used as a diagnostic marker and has become a standard tool in the serological assessment of chronic HBV infection [21,22,23]. Over the years, several commercial methods have been developed to detect HBsAg, including automated immunoassays such as chemiluminescent microparticle immunoassay (CMIA) and the latest generation of highly sensitive enzyme-linked immunosorbent assay (ELISA) [24,25]. These assays, which utilize monoclonal antibodies binding to HBsAg as the core component, are widely used for rapid screening and detection of HBsAg in large sample numbers and are relatively cost-effective. To date, however, the application of variable nanobodies in the detection of HBsAg has not been explored. This presents a potential opportunity for the development of a novel diagnostic approach for HBV infection.

In this study, we aimed to develop a novel approach for the detection of HBsAg by taking advantage of VNARs from the whitespotted bamboo shark (*Chiloscyllium plagiosum*). We first immunized the sharks with HBsAg protein and then isolated total RNA from peripheral blood leukocytes (PBLs). An anti-HBsAg VNAR-displaying phage library was subsequently constructed and subjected to multiple rounds of phage display panning against HBsAg. This resulted in the isolation of 20 specific VNARs against HBsAg. Three of these VNARs were selected and expressed in *E. coli* WK6 cells, and their specificity and binding potency to HBsAg were determined. The binding ability of these VNARs to HBsAg was then evaluated pairwisely using a sandwich ELISA, demonstrating their potential for more efficient detection of HBsAg. These findings lay the foundation for future studies aimed at developing a novel and efficient approach for detecting HBsAg in a range of biological samples.

## 2. Results

### 2.1. Isolation of Specific Nanobodies against HBsAg

After immunization of whitespotted bamboo sharks with HBsAg, the total RNA was isolated from PBLs. The gene fragments encoding VNARs were amplified by PCR using reverse-transcribed cDNA as the template. As expected, the PCR product identified by agarose gel electrophoresis was approximately 400 bp. The VNAR gene fragments were inserted into pComb3XSS vectors and electroporated into *E. coli* TG1 cells to construct an anti-HBsAg VNAR phage display library. The capacity of this library was about 1 × 10^9^ colony-forming units (CFU). The 48 colonies were randomly selected, and colony PCR was performed to verify the insertion rate using specific primers. As shown in Figure 1a, a 100% insertion rate of the VNAR phage display library was identified in these 48 colonies. In general, we successfully constructed an anti-HBsAg VNAR phage display library with high quality.

After the establishment of the immune nanobody library, we isolated single-domain antibodies against HBsAg by bio-panning. After two rounds of panning, the ratio of enrichment has increased from 7 to 212 (Figure 1b). Then we randomly picked 96 colonies and tested whether these VNARs could bind HbsAg through phage ELISA assay. 93.75% of colonies (90 in 96) showed a high affinity with HbsAg (Figure 1c). The 20 different VNAR sequences from these positive colonies were further identified by sequencing (Figure 1d). These sequences were named from HB1 to HB20. After comparison and analysis, we further divided these nanobody sequences into four families according to the location of cysteine in the CDRs, which all belonged to type II of VNAR. 

### 2.2. Expression of Soluble Single Domain Antibodies and Binding Potency Assay

In order to characterize the binding potency of the VNAR single-domain antibodies, five typical VNARs were expressed in *E. coli* WK6 cells and purified by Ni-NTA resin: HB1, HB3, HB14, HB17, and HB18. The analysis of sodium dodecyl sulfate-polyacrylamide gel electrophoresis (SDS-PAGE) showed that the purity of single-domain antibodies was more than 90% (Figure 2a). ELISA was performed to measure the specificity and binding potency of VNAR with antigen. A panel of proteins available in the laboratory, including tag proteins (GFP, hIgG Fc, and GST), bovine serum albumin, and human serum albumin, was used for specificity testing. There was no binding signal detected with irrelevant proteins, while a strong signal was obtained with HBsAg (Figure 2b). Further nanobodies gradient testing showed that the EC_50_ values of HB14, HB17, and HB18 were 4.864 nM, 4.260 nM, and 8.979 nM, respectively (Figure 2b–d). These results indicated that our nanobodies had a high binding capability with HBsAg. 

### 2.3. Testing of HBsAg-Specific VNARs Available for Sandwich ELISA 

We next performed the sandwich ELISA assay using HB14, HB17, and HB18 nanobodies pairwisely to determine if these nanobodies can simultaneously interact with HBsAg. As shown in Figure 3, each group showed a high binding affinity with HBsAg. The highest absorbance occurred in the HB18 (capture antibody) and HB14 (detection antibody) group, at about 6.38 fold compared with the control. The HB17 (capture antibody) and HB18 (detection antibody) groups, showed the lowest absorbance at about 3.56 fold compared with the control. There was no signal detected when the same nanobody was used as the capture antibody and detection antibody. These results demonstrated that HB14, HB17, and HB18 could bind HBsAg at different epitopes.

## 3. Discussion

Antibodies have always played important roles in the development of diagnostic technologies. Polyclonal antibodies, monoclonal antibodies, and genetically engineered antibodies have each represented significant milestones in this field’s evolution. The ELISA technique is currently widely used in an immunoassay for the detection of viruses, attributed to its advantages of being simple, fast, practical, cost-effective, and having low requirements for experimental conditions and personnel [26,27]. Although monoclonal antibodies have become the primary detection reagents used in ELISA, their large size (~150 kDa) and long production cycle limit their development and application [28]. VNARs have emerged as a promising alternative due to their low molecular size (~12 kDa) and simple gene structure, making them easier to produce using bacteria and fuse with reporter genes [11,29]. Noteworthy, research has indicated that antibodies were particularly stable in sharks due to the high concentration of urea in their blood, which could make them more stable than the antibodies found in organisms without such urea enrichment [5]. Therefore, a phage display anti-HBsAg VNAR library was constructed by immunizing whitespotted bamboo sharks with HBsAg, and 20 specific nanobodies were screened. The library’s excellent capacity enabled the screening of a larger number of HBsAg-specific sequences to fulfill the demand for antibodies with different characteristics. 

We further produced six of these specific nanobodies using the *E. coli* expression system and purified recombinant proteins by osmotic shock method. The EC_50_ values of these antibodies with HBsAg were in the single-digit nanomolar range, indicating that their binding potency to HBsAg is relatively strong, which is essential for the development of an antibody in a diagnostic test. It is well known that since the detection of a target requires the binding of two different antibodies, sandwich ELISA typically shows high specificity and great confidence in the results [30]. Inspired by this, we investigated the possibility of simultaneous binding of these three nanobodies to antigens. Our results demonstrated that the three nanobodies, HB14, HB17, and HB18, have different antigen binding sites on HBsAg, which means that they can be applied in sandwich ELISA diagnosis. These three antibodies offer a new possibility for sensitive and accurate diagnosis of HBsAg in biological samples.

In summary, the construction of an anti-HBsAg VNAR library and the screening of 20 specific nanobodies provide a promising novel method for the detection of HBsAg. This method has a simpler preparation process, lower cost, and higher sensitivity than conventional detection methods, making it an ideal diagnostic tool for resource-limited environments. 

## 4. Materials and Methods

### 4.1. Ethics Statement

All procedures were approved by the Animal Care and Use Committee of the Third Institute of Oceanography, Ministry of Natural Resources, and conformed to the guidelines of the Fujian Provincial Department of Science and Technology for the Administration of Affairs Concerning Experimental Animals. 

### 4.2. Immunization of Whitespotted Bamboo Sharks

The whitespotted bamboo sharks, weighing about 0.5 kg, were kept in seawater at approximately 28 °C in large indoor pools at the Third Institute of Oceanography, Ministry of Natural Resources. 

Prior to blood collection and immunization, sharks were anesthetized in artificial seawater using MS-222 (Sigma, St. Louis, MO, USA) at a concentration of approximately 0.1% (*w*/*v*). The immunization protocol was modified from the previous description [31,32]. In brief, we immunized two whitespotted bamboo sharks four times, and the interval between immunizations was 4 weeks. 200 μg HBsAg dissolved in elasmobranch-modified PBS and mixed with or without equal amounts of adjuvant. The complete Freund’s adjuvant (Sigma, St. Louis, MO, USA) used in the first immunization and incomplete Freund’s adjuvant (Sigma, St. Louis, MO, USA) used in the second immunization were subcutaneously injected into the pectoral fin. In the latter two immunizations, 100 μg soluble HBsAg was directly injected into the caudal vein. The shark PBLs were isolated to construct an immune VNAR library two weeks after the fourth immunization. We phlebotomized from the caudal vein. The porcine heparin sodium salt was added to prevent the blood from clotting. Blood plasma was isolated by spinning at 2000 rpm for 10 min. The 1 mL of TRIZOL (Thermo Fisher Scientific, Cleveland, OH, USA) was added to every 200 μL of hematocyte and stored in the −80 °C refrigerator. 

### 4.3. VNAR Library Construction

Total RNA was isolated from the purified PBLs using TRIZOL, according to the manufacturer’s instructions. Then we used PrimeScript™ II 1st Strand cDNA Synthesis Kit (Takara, Beijing, China) to synthesize cDNA. To obtain the VNAR genes, PCR was then performed with VNAR-specific primers, which were as previously described [33]. The VNAR genes of the first PCR products were analyzed by agarose gel electrophoresis. The second round PCR products were digested with *Sfi*I restriction enzyme (New England BioLabs, Ipswich, MA, USA) and inserted into the phagemid pComb3XSS with T4 ligase (Thermo Scientific, Waltham, MA, USA). After purification, the ligation products were transformed into *E. coli* TG1 cells by electroporation. The cells were plated onto 2 × YT agar medium (5 g/L NaCl, 16 g/L tryptone, 10 g/L yeast extract, and 15 g/L agar) containing 100 µg/mL ampicillin and cultured at 30 °C for 16 h. The cells were also serially diluted from 10^6^ to 10^10^, and the library size was determined by counting the colony number. 

### 4.4. Selection of HBsAg-Specific VNAR

The immuno-tube (Nunc, Wiesbaden, Germany) was coated with 2 mL of HBsAg. In each round of selection, the antigen concentration used for immobilization was gradually decreased: 50 µM for round 1 and 25 µM for round 2. The immuno-tube filled with 5% MPBS (5% non-fat powdered milk in PBS) was used as a control. Both immuno-tubes were incubated at 4 °C overnight. After washing and blocking with 5% MPBS at room temperature for 1 h, 1 mL of amplified phage display library (1 × 10^13^ pfu/mL) was added to each immuno-tube and incubated for 1 h at room temperature on a rotator. The dissociated phage was removed by washing 5 times with 0.1% PBST (PBS containing 0.1% Tween-20) in Round 1. In subsequent panning, the number of washes increased to 10. After washing, the bound phage was eluted with 100 mM of HCl at room temperature for 7 min on a rotator, and the eluent was neutralized with 1 M Tris buffer (pH 7.4). The eluated phage was added to 10 mL of exponentially growing *E. coli* TG1 cells (OD600 of 0.5). After incubating the eluate-*E. coli* cells mixture at 37 °C on a shaker, plated cells onto 2 × YT agar medium containing 1% glucose and 100 μg/mL ampicillin, and then incubated at 37 °C overnight. The cells on the plate were then collected, followed by superinfection of the helper phage in 100 mL of the 2 × YT medium. After 16-h shaking at 25 °C, the phage was collected and purified for the next round of selection.

### 4.5. Phage ELISA Assay

The anti-HBsAg phage library was plated on a 2 × YT agar plate containing 100 μg/mL ampicillin to isolate the single colonies. Single colonies were individually inoculated into 1 mL of 2 × YT medium supplemented with 100 μg/mL of ampicillin and 10^11^ pfu/mL of the M13KO7 helper phage in a 96 deep-well plate, and then the medium was shaken at 37 °C for 1 h. The culture was grown at 25 °C overnight with shaking. The bacteria were deposited by centrifugation at 4000 rpm for 20 min. The phage supernatant was collected.

Each well of the 96-well plate (NEST, Wuxi, China) was coated with 100 ng HBsAg in PBS buffer at 4 °C overnight. 5% non-fat powdered milk in PBS buffer was used as the control. After the plate was blocked with 5% MPBS at 37 °C for 2 h, 100 μL phage supernatant was added to the plate and incubated for another 2 h. The HRP conjugated mouse anti-M13 antibody (Sino Biological, Beijing, China), diluted at 1: 20,000 in 100 μL 5% MPBS, was added to the plate and incubated for 1 h at room temperature. After 9 washes with 0.1% PBST, binding was detected by TMB Single-Component Substrate solution (Solarbio, Beijing, China), and the reaction was quenched by 200 mM Sulfuric acid. The signals were quantified by OD450 using a plate reader (Varioskan^TM^ LUX, Thermo Scientific, Waltham, MA, USA). A positive binding was identified when the cutoff value was 5-fold higher than that of the control, then these VNAR sequences were identified by DNA sequencing. 

### 4.6. Expression and Purification of VNARs

The pComb3XSS phagemids containing the positive VNAR gene were transformed into *E. coli* WK6 cells by heat shock at 42 °C for 90 s. The colonies were inoculated into 200 mL TB medium (24 g/L tryptone, 12 g/L yeast extract, 9.4 g/L K_2_HPO_4_, 2.2 g/L KH_2_PO_4_, 4 mL/L glycerol) with 100 μg/mL of ampicillin and shook at 37 °C until OD600 reached 0.8–1.0. The culture medium was then replaced with a TB medium containing 1 mM IPTG, 100 μg/mL ampicillin, and shaken at 28 °C overnight for soluble protein production. Cells collected by centrifuging were suspended in 6 mL of ice-cold TES buffer (0.2 M Tris-HCl pH 8.0, 0.5 mM EDTA, 20% sucrose). The cell suspension was incubated on ice for 1 h. Cells were lysed by adding 12 mL of 1/4 TES buffer (TES diluted at 1:3 with ultra-pure water) and further incubated for 45 min on ice. The slurry was then centrifuged at 10,000 rpm for 30 min at 4 °C. Soluble VNARs containing His-tag were purified from the cell lysate by Ni-NTA resin (TransGen Biotech, Beijing, China).

### 4.7. Binding Specificity Assay for VNARs

The specificity of VNARs was evaluated by ELISA. Some antigens: BSA, HSA, GFP, hIgG Fc, and GST, were coated on a 96-well plate at 1 μg/mL in PBS buffer, 100 μL/well, at 4 °C overnight, and 5% MPBS was as control. After blocking with 5% MPBS at 37 °C for 2 h, The VNARs in 2% MPBS were added to the plate at the concentration of 100 ng/μL and incubated at room temperature for 2 h. Followed by 9 washes with 0.1% PBST; mouse His-tag mAb (Abclonal, Wuhan, China) was diluted at 1:2000 with 1% MPBS and added. The 9 washes were performed after incubating for 2 h at room temperature, and the goat anti-mouse IgG (Abclonal, Wuhan, China) was diluted at 1:5000 with 1% MPBS and incubated for 1 h at room temperature. After 9 washes, binding was detected by adding 100 μL TMB Single-Component Substrate solution. 200 mM Sulfuric acid was added to inhibit the reaction, and the signals of OD450 were detected by a plate reader. The data were processed by Prism 7 for analysis.

### 4.8. ELISA Binding Potency Assay for VNARs

The purified VNARs were used to test binding potency with HBsAg by ELISA. The HBsAg was coated on a 96-well plate at 1 μg/mL in PBS buffer, 100 μL/well, at 4 °C overnight. At the same time, 5% MPBS was as control. After blocking with 5% MPBS at 37 °C for 2 h, The VNARs were serially 10-fold dilution and added to the plate at the concentration of 1 μM to 10 pM in the 2% MPBS and incubated at room temperature for 2 h. Followed by 9 washes with 0.1% PBST, mouse His-tag mAb was diluted at 1:2000 with 1% MPBS and added. There were 9 washes after incubating for 2 h at room temperature. The goat anti-mouse IgG was diluted at 1: 5000 with 1% MPBS and incubated for 1 h at room temperature. After 9 washes, binding was detected by adding TMB Single-Component Substrate solution as the mentioned steps above. The signals were quantified by OD450 using a plate reader. The data were processed by Prism 7 to analyze the concentration for EC_50_.

### 4.9. Sandwich ELISA for VNARs 

Sandwich ELISA was adopted to test the possibility of both nanobodies binding to antigens at the same time. The purified HB14, HB17, and HB18 were coated on the plate with 3 parallel wells at 1 μg/mL in PBS buffer, 100 μL/well, and incubated at 4 °C overnight. Prior to all incubation steps, 3 washes were performed by PBST. The next day, 5% MPBS was incubated at 37 °C for 2 h. Then 100 μL HBsAg diluted at 1 ng/μL was added, and the plate was placed on a shaker at room temperature for 2 h. Meanwhile, the PBS buffer was used as the blank group. The phage of HBsAg-specific nanobodies (preparation method mentioned above) was incubated on a shaker for 2 h as well. Finally, the HRP-conjugated mouse anti-M13 antibody was used to detect the binding signals according to the steps above.

## Figures and Tables

**Figure 1 marinedrugs-21-00237-f001:**
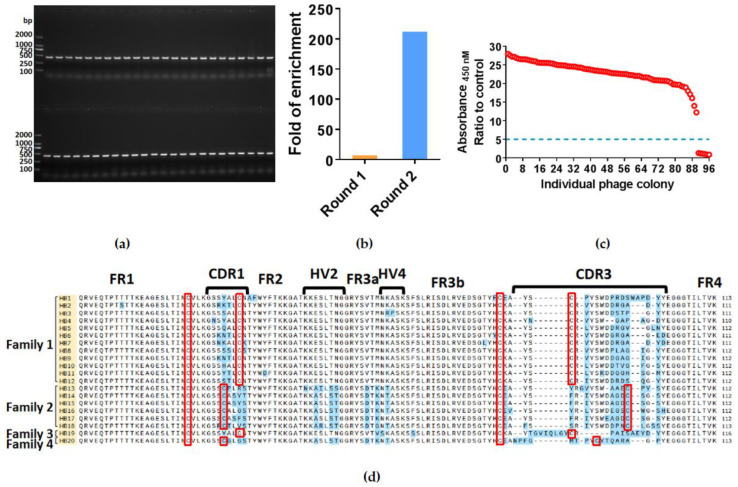
Phage display panning of HBsAg-specific VNARs. (**a**) PCR verified the quality of the anti-HBsAg phage display library. (**b**) The enrichment ratio of each panning round. (**c**) Phage ELISA results for 96 single colonies. The individual colonies randomly picked from second-round panning are marked with red circles, while the demarcation line between positive and negative colonies is denoted by a blue dashed line. (**d**) The amino acid sequences of screened VNARs. FR: Framework region, CDR: Complementarity-determining region, HV: Hypervariable region. The orange highlight indicates the name of sequences, and the blue highlight indicates the amino acids not conserved in these VNAR sequences. The Cysteine is shown in a red box.

**Figure 2 marinedrugs-21-00237-f002:**
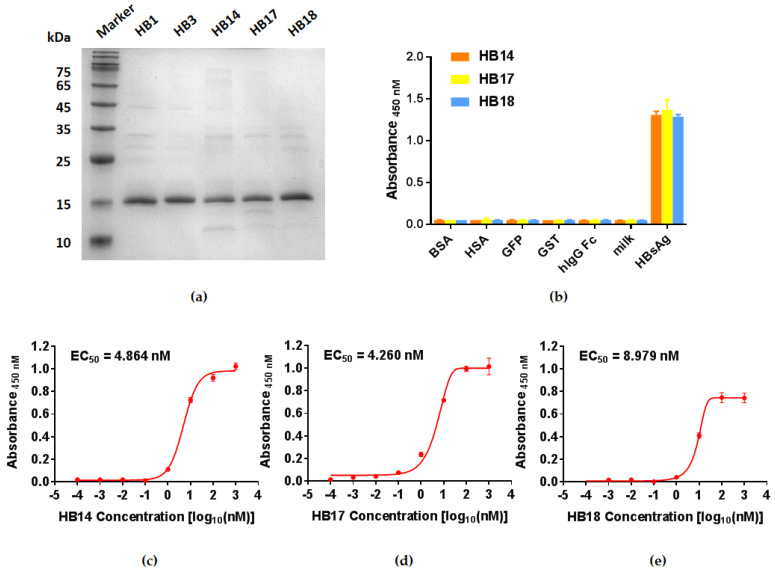
HBsAg-specific VNARs: HB1, HB3, HB14, HB17, and HB18, and the binging potency testing. (**a**) SDS-PAGE of anti-HBsAg single domain antibodies: HB1, HB3, HB14, HB17, and HB18. (**b**) Specificity detection of HB14, HB17, and HB18, measured by ELISA. (**c**–**e**) Binding potency of HB14, HB17, and HB18 against HBsAg, measured by ELISA.

**Figure 3 marinedrugs-21-00237-f003:**
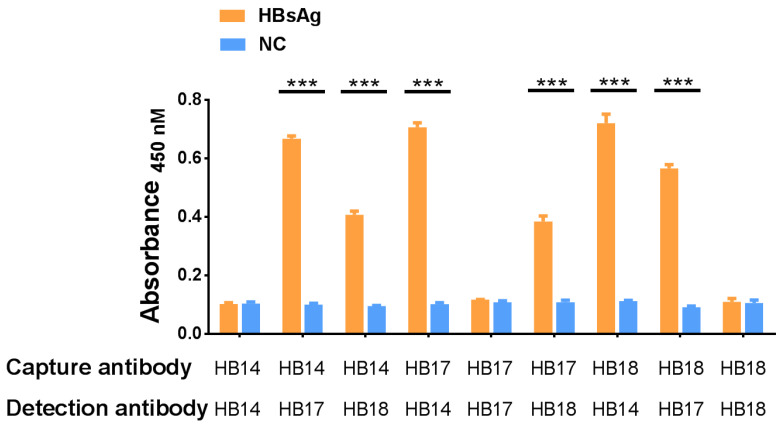
Sandwich ELISA detection of different epitopes of antigens bound by two nanobodies simultaneously. (The orange column indicates the use of HBsAg in combination with two nanobodies, and the blue column indicates PBS instead). Stars indicate statistical significance between the experiment and control (*n* = 3, *** *p* < 0.001).

## Data Availability

The data presented in this study are available on request from the corresponding author.

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
