# Peer review of "Isolation and Characterization of Targeting-HBsAg VNAR Single Domain Antibodies from Whitespotted Bamboo Sharks (Chiloscyllium plagiosum)"

_marinedrugs, 2023, doi:10.3390/md21040237_

Round 1

Reviewer 1 Report

In this study, the whitespotted bamboo sharks (Chiloscyllium plagiosum) were immunized with the recombinant HBsAg protein. their results provide a new possibility for the application of  VNAR in HBV diagnosis and
also demonstrate the feasibility of using VNAR for medical testing.

-The English language should be improved.

-You should determine the affinity of nanobodies by BioCore (SPR thechnology)

_You must evaluate specificity and sensitivity of nanobodies.

Author Response

Enclosed please find the "Response to Reviewer" file

Reviewer 2 Report

The manuscript of X. Jiang et al. describes the isolation of shark single-domain antibodies binding to the hepatitis B surface antigen in vitro. Using standard protocols, the authors have been able to isolate 3 different single-domain binders that react with the antigen with high affinity. The experiments presented allowed to characterize these new biomolecules in detail (DNA sequence, protein profile, binding properties). I have no major criticism about the described work, but for me the two points listed below should be dealt to render this manuscript even more interesting.

1.     Sandwich ELISA (Figure 3). It is unclear for me why the authors did not test what is happening when the same single-domain antibody was used as capture and as detection antibody. That represents an important control for verifying that the quality of HBsAg used in this assay is adequate for stating that the 3 single-domain recognize different epitopes. Indeed, if no signal is obtained in this case, it can be concluded what is written in lines 146-147. However, I expect that the HBsAg preparation contains likely aggregates and that these aggregates upon capture can be detected with the same antibody used for capture. We experienced this situation several times. If the quality of the antigen is not properly controlled (i.e. in monomeric form) all the biomolecules described in Figure 3 can bind to the same epitope. To be clear, the method used by the authors for the epitope mapping/binning can only be considered as reliable if the antigen is monomeric. This should be verified. In addition, there’s no comment in the text about the origin and the protein quality of the antigen used for the selection of the phage particles and the ELISAs.

2.     Advance for the diagnostic of HBV. Although the work done allowed to isolate new biomolecules to HBsAg, I did not well understand what would be the advantage of these tools for the diagnosis of HBsAg present in biological samples. Conventional antibodies (described in numerous publications in Pubmed) are also easy to prepare and of high affinity and specificity in general. The authors should propose an experiment or an application which clearly indicates that the isolated single-domain antibodies are of higher value than conventional antibodies (sensitivity of the test, robustness...). In my experience, using phage particles as detecting antibodies, as proposed by the authors as an alternative to conventional antibodies, leads generally to higher background when blood or other biological samples are used for performing the ELISA test. If this point is considered, I would be convinced about the importance of publishing this work.

Author Response

Enclosed please find the "Response to Reviewer" file. Thank you. 
